# Drug Release via Ultrasound-Activated Nanocarriers for Cancer Treatment: A Review

**DOI:** 10.3390/pharmaceutics16111383

**Published:** 2024-10-27

**Authors:** Khaled Armouch Al Refaai, Nour A. AlSawaftah, Waad Abuwatfa, Ghaleb A. Husseini

**Affiliations:** 1Biomedical Engineering Program, College of Engineering, American University of Sharjah, Sharjah P.O. Box 26666, United Arab Emirates; b00086992@aus.edu; 2Materials Science and Engineering Ph.D. Program, College of Arts and Sciences, American University of Sharjah, Sharjah P.O. Box 26666, United Arab Emirates; g00051790@alumni.aus.edu (N.A.A.); g00062257@alumni.aus.edu (W.A.); 3Department of Chemical and Biological Engineering, College of Engineering, American University of Sharjah, Sharjah P.O. Box 26666, United Arab Emirates; 4Biosciences and Bioengineering Ph.D. Program, College of Engineering, American University of Sharjah, Sharjah P.O. Box 26666, United Arab Emirates

**Keywords:** nanocarriers, ultrasound, chemotherapy, ultrasound-induced drug release, nanomedicine, targeted drug delivery

## Abstract

Conventional cancer chemotherapy often struggles with safely and effectively delivering anticancer therapeutics to target tissues, frequently leading to dose-limiting toxicity and suboptimal therapeutic outcomes. This has created a need for novel therapies that offer greater efficacy, enhanced safety, and improved toxicological profiles. Nanocarriers are nanosized particles specifically designed to enhance the selectivity and effectiveness of chemotherapy drugs while reducing their toxicity. A subset of drug delivery systems utilizes stimuli-responsive nanocarriers, which enable on-demand drug release, prevent premature release, and offer spatial and temporal control over drug delivery. These stimuli can be internal (such as pH and enzymes) or external (such as ultrasound, magnetic fields, and light). This review focuses on the mechanics of ultrasound-induced drug delivery and the various nanocarriers used in conjunction with ultrasound. It will also provide a comprehensive overview of key aspects related to ultrasound-induced drug delivery, including ultrasound parameters and the biological effects of ultrasound waves.

## 1. Introduction

It is estimated that in 2023, approximately 1,958,310 new cases of cancer and 609,820 cancer deaths occurred in the United States [1]. It encompasses a wide range of diseases that can affect any organ when abnormal cells multiply uncontrollably and spread to other organs. Cancer rates continue to rise worldwide, imposing a significant physical, psychological, and financial burden on individuals, families, communities, and healthcare systems [2]. Due to the high prevalence and mortality rate of cancer, more effective methods of prevention and treatment are urgently needed. Early detection of cancerous tumors allows for successfully treating about one-third of patients using local therapies such as surgery or radiotherapy [3]. However, a systemic approach involving chemotherapy is necessary for the remaining patients for effective cancer management [3].

Conventional chemotherapy is still used to stop cancer cells from spreading throughout the patient’s body, but the drugs are not very selective and kill both cancerous and healthy cells. As a result, the body’s healthy cells suffer from poor performance and unwanted side effects like fatigue, hair loss, nausea, vomiting, and decreased appetite [4]. These adverse reactions harm the patient’s quality of life and can affect the course of the treatment. Recent studies have led to the development of drug delivery systems utilizing therapeutics based on nanocarriers, which have several benefits, such as small size, extended circulation time, and systemic stability [5]. With the use of nanocarriers, these drug delivery methods seek to reduce the systemic toxicity of administered drugs or nanocarriers while delivering a constant concentration to the intended site [6]. To further improve their therapeutic efficacy, a plethora of nanomaterials sensitive to endogenous and exogenous stimuli have been created [5].

These materials provide precise control over the spatial and temporal release of the drug payload. In particular, using stimulus-responsive nanocarriers in cancer treatments has improved the specificity of chemotherapy drugs and enhanced treatment localization. This approach enables the differentiation between cancerous and healthy cells, triggering the release of the nanocarrier’s payload directly at the target site.

Several systems that use physical stimuli, such as light, magnetic fields, ultrasound, x-rays, and hyperthermia, are being developed to trigger the localized release of anticancer drugs at the tumor site [7]. An ideal triggering strategy would exclusively deliver precise energy to the cancerous tissue. Each approach currently under investigation has its advantages and disadvantages, as shown in Table 1. For instance, applying light and magnetic fields as triggering mechanisms is safe, but can only be used on surface tumors. On the other hand, using X-rays as the physical stimulus can cause ionization in healthy cells but offers better tissue penetration. As a drug delivery system triggering mechanism, ultrasound has several advantages. It combines the advantages of non-invasive, non-ionizing radiation with high temporal and spatial precision and potential imaging utility.

This review will focus on the advancements and applications of ultrasound-activated drug delivery systems. Ultrasound techniques are widely utilized in the healthcare sector due to their non-invasive nature, ease of use, affordability, and simplicity. Common applications include imaging, diagnostics, object detection, and distance measurement [8]. Different effects can be obtained by adjusting the frequency of ultrasound waves, including control of drug delivery and cell function [9]. Therefore, ultrasound techniques act as a non-invasive triggering mechanism for drug delivery, enabling precise control over drug release at the targeted location.

## 2. Physics of Ultrasound Waves

Acoustic waves with frequencies higher than 20 kHz are called ultrasound waves. These are longitudinal mechanical waves that propagate through a medium via pressure variations. Ultrasound waves exhibit various physical characteristics, including attenuation, reflection, refraction, amplification, absorption, and scattering. Unlike MRI and CT scans, ultrasound technology can be brought directly to the patient, making it convenient. It is also widely used in the medical field for both therapeutic and diagnostic purposes.

In diagnostic sonography, transducers typically generate ultrasound waves that contain piezoelectric crystals. Like all piezoelectric materials, these crystals can convert electrical signals into mechanical pressure waves and vice versa. Researchers have utilized these ultrasound transducers to target specific regions, facilitating ultrasound-assisted drug release while simultaneously monitoring the site for potential additional treatments. As sound waves travel through a material, they create a series of compression and decompression events that alter the medium’s local density. Consequently, when ultrasound waves pass through a liquid medium, the dissolved gas nuclei within the liquid expand and collapse due to mechanical vibrations. The primary mechanisms through which ultrasound enhances targeted drug delivery are the physical effects generated by the oscillation and implosion of these cavitation bubbles.

It is possible to modulate ultrasound waves by adjusting various parameters, like frequency and intensity [10]. While intensity determines the amount of energy delivered to the desired location, frequency allows modulation of parameters like potential cavitation and penetration depth [10]. It is also possible to adjust the duty cycle, which is the sequence and duration of ultrasound pulses, and the exposure time frame. Lastly, specific results can also be obtained by applying focused or unfocused ultrasonic waves.

### 2.1. Ultrasound Parameters

#### 2.1.1. Frequency

The frequency of an ultrasound wave is the most widely used parameter to characterize it. Equation (1) illustrates how the frequency is defined as the ratio of the wave’s speed (*c*) to wavelength (*λ*). It is possible to apply a broad spectrum of ultrasonic frequencies to human tissue. These fall into three categories: (1) Medium Frequency Ultrasound (0.7–3 MHz), (2) Low-frequency Ultrasound (20–200 kHz), and (3) High-frequency Ultrasound (>3 MHz) [10]. The ultrasonic wave’s frequency parameter affects two factors: (1) tissue penetration depth and (2) spatial resolution [10].
(1)$$f=\frac{c}{\lambda }$$

A high-frequency ultrasound wave provides better resolution but has a lower tissue penetration depth [10]. Conversely, tiny details will go unnoticed at low frequencies and only large objects will be identified [10]. However, since frequency and penetration depth are inversely correlated, low frequencies will yield better penetration depth [10]. Therefore, for diagnostic imaging, a higher ultrasound frequency should be chosen.

#### 2.1.2. Intensity and Exposure Duration

Ultrasound Intensity (I), expressed in W/cm^2^, is the power the ultrasound wave carries over the surface area it is applied to [10]. Thus, as indicated by Equation (2), the ultrasonic intensity can be correlated with the applied acoustic pressure (*P*), the density of the medium to which it is applied (*ρ*), and the speed of ultrasonic wave propagation within the medium (*c*) [10]. There are two categories into which the intensities of ultrasound waves can be separated: (1) Low-Intensity Ultrasound (0.125–3 W/cm^2^) and (2) High-Intensity Ultrasound (>3 W/cm^2^) [10]. When ultrasonic waves of low intensity pass through biological tissue, they cause reversible changes without causing any damage, whereas ultrasound waves of high intensity cause hyperthermia and other biological effects that cause irreversible changes in the tissue.
(2)$$I=\frac{P^{2}}{\rho c}$$

Ultrasound waves can be applied either continuously or in pulses. Continuous ultrasound produces thermal effects by modifying cell membrane permeability, promoting intracellular calcium, and inducing tissue regeneration [11]. When ultrasonic waves are applied discontinuously as a series of pulses with a bit of downtime in between, pulsed US (PUS) primarily results in non-thermal effects that enhance fibrous tissue extensibility, raise the pain threshold, and increase tissue metabolism [11]. The pulse sequence comprises pulses that are periodically repeated at a specific frequency; this frequency is commonly expressed in repetitions per second, or Hertz (Hz). The adequate ultrasound exposure time is represented by and corresponds to the duty cycle, which is determined by the frequency of pulse repetition and the duration of each pulse.

Various techniques are employed to measure the ultrasound wave’s intensity when discontinuous ultrasound waves are used. These methods rely on the pulse span and shape of the ultrasonic beam, representing the temporal and spatial variation that influences the measurement of the ultrasonic wave’s intensity.

An ultrasound transducer generates single pulses, each of which is comprised of multiple cycles, as shown in Figure 1, which results in intensity variations within the pulse. Three measurements can be used to characterize these temporal variations: (1) the temporal peak, (2) the pulse average, and (3) the temporal average. When the ultrasonic wave pulse reaches its maximum amplitude, a measurement known as the temporal peak is taken, which characterizes the maximum intensity attained during a single pulse. The ultrasound wave’s average intensity over the course of a single pulse is known as the pulse average measure. In the temporal average measurement, the ultrasonic wave’s intensity is averaged over the pulse repetition period. As a result, this measure records the lowest intensity since it captures both the intensity during the pulse’s application and the period of time between that pulse and the next in the sequence during which no intensity is present.

It is crucial to remember that the sound beam is not uniformly shaped and sized along its whole length when using ultrasonic waves. Consideration must be given to the spatial variations that result from this issue. The far zone, the focal zone, and the near zone make up the three sections of the sound beam, as shown in Figure 2. The sound beam narrows as it travels farther from the transducer, where it originated, and eventually reaches the focal zone, where it is the narrowest. The area of the sound beam that comes before the focal zone is called the near zone, and the area that comes after the focal zone, where the sound beam begins to widen, is called the far zone. The most considerable intensity produced by the sound beam is in the focal zone where the pressure created by the ultrasonic wave is concentrated into the smallest attainable area. The transducer’s piezoelectrical crystal diameter and the ultrasound wave’s frequency dictate the dimensions and configuration of these zones. These spatial variations are described by two measurements: (1) the spatial average intensity and (2) the spatial peak intensity. While the spatial average intensity is measured at the center of the sound beam close to the transducer and represents the average intensity generated across the entire area covered by the sound beam, the spatial peak intensity is observed at the narrowest area of the beam, the focal zone, where the largest intensity is generated. The following six combinations can be used to measure intensity using both the temporal and spatial variation of the ultrasound waves:SPTP (Spatial Peak Temporal Peak Intensity) is the highest intensity measured at the sound beam’s focal zone during an ultrasound peak pulse.SPTA (Spatial Peak Temporal Average Intensity) measures the average intensity at the sound beam’s focal zone over a pulse repetition period. It corresponds to the thermal effect of ultrasound waves.SPPA (Spatial Peak Pulse Average Intensity) measures the average intensity at the sound beam’s focal zone over a pulse’s duration. It is connected to the ultrasonic waves’ mechanical and cavitation effects.SATP (Spatial Average Temporal Peak Intensity) is the highest intensity measured during an ultrasound pulse, averaged across the entire sound beam.SATA (Spatial Average Temporal Average Intensity) measures the averaged ultrasound wave intensity over a pulse repetition period and across the entire sound beam.SAPA (Spatial Average Pulse Average Intensity) measures the averaged intensity of ultrasound waves throughout a pulse and across the entire sound beam.

#### 2.1.3. Focused and Unfocused Ultrasound

Ultrasound waves can be applied using focusing or non-focusing transducers; each transducer type generates a distinct sound beam. Typically, sonophoresis, a technique that improves transdermal delivery and the physical effects of the ultrasound wave, is accomplished with non-focusing ultrasound transducers. On the other hand, focused ultrasound transducers concentrate ultrasonic radiation into a small area, increasing the intensity at that area.

This type of focused ultrasound is known as high-intensity focused ultrasound, or HIFU, when the intensity is higher than 5 W/cm^2^. By focusing ultrasound waves, a significant amount of energy can be delivered to a small area (referred to as the focal zone) at a particular depth within the body while keeping the other areas where the ultrasound wave propagates through safe and without any damage. Researchers have used other imaging modalities, like CT, MRI, and diagnostic ultrasound, to direct the HIFU sound beam toward particular organs, tissues, and tumors. HIFU ablation is a common HIFU application that involves localized heating at the ultrasound beam’s focal point to selectively destroy tissue. This method has been applied in place of surgery to remove cancerous tumors.

Low Intensity Focused Ultrasound, also known as LIFU, is the second type of focused ultrasound, which appears when the intensity is between 0.125 and 3 W/cm^2^ [10]. Contrary to HIFU, LIFU has been utilized in preclinical trials to enhance drug control and release from stimuli-triggered nanoparticles since it is not associated with a significant energy accumulation at the focal point.

## 3. Biological Ultrasound Effects

### 3.1. Thermal Effects

When an ultrasound wave travels through a medium, it loses energy due to absorption and scattering, and its kinetic energy is transformed into thermal energy when it is absorbed. With the assumption that the initial ultrasound wave had a frequency of *f* and an initial pressure amplitude of *P*_0_, the attenuated pressure amplitude *P_x_* at point *x* can be found using Equation (4).
(3)$$P_{x}=P_{0}e^{-ax}$$

The absorption coefficient (*α*) determines the amount of energy absorbed. As indicated by the following equation, this coefficient depends on the ultrasonic wave’s frequency *f*, a reference absorption coefficient *α*_0_, and a constant *n*.
(4)$$\alpha =\alpha_{0}f^{n}$$

As Equation (4) illustrates, a greater amount of the ultrasonic kinetic energy will be absorbed by the target tissue and transformed into thermal energy when the ultrasonic wave frequency increases. The affected tissue will begin to burn and undergo necrosis from protein denaturalization if the temperature rises to a point where local ultrasound-induced hyperthermia occurs. The sensitivity of various tissue types to this increase in temperature will vary. Additionally, the tissue type will affect the time needed for protein denaturation to occur. The transition temperature between mild and strong hyperthermia has been identified as 43 °C. Mild hyperthermia occurs if the temperature is kept below 43 °C while temperatures higher than 43 °C will result in strong hyperthermia.

#### 3.1.1. Mild Hyperthermia

The heat generated during mild hyperthermia causes the blood to dilate and the vascular walls’ permeability to increase, which in turn causes an increase in blood flow [13]. Even during prolonged episodes of mild hyperthermia, no protein denaturation occurs at these temperatures [10]. Both malignant and healthy tissue see an increase in blood flow; however, the effect appears to be lessened in malignant tumors [14]. On the other hand, it seems that cancer cells are more vulnerable to the negative effects of heat. Hyperthermia causes protein denaturation and pathway changes in cancer cells, making them more susceptible to irradiation and chemotherapy [15]. Furthermore, thermosensitive nanocarriers are designed to release their payload in response to mild hyperthermia [16]. Since mild hyperthermia increases vascular permeability and blood flow, it can also be used to enhance nanoparticle accumulation at tumor sites [17,18].

The acoustic pressure should be moderated, and the frequency should be kept in the MHz range to achieve controlled mild hyperthermia [10]. By lengthening the intervals between each pulse in the sequence, the duty cycle of the ultrasonic wave can also be utilized to regulate overheating incidents that may arise [10].

#### 3.1.2. Strong Hyperthermia

Strong or high hyperthermia, above 43 °C, is used to induce protein denaturation, breakdown of tumor vasculature, and rapid cell necrosis [10]. When combined with high intensity focused ultrasound (HIFU), it raises the local temperature at the focal point to between 50 and 80 °C, which causes cellular proteins to rapidly coagulate due to heat and causes irreversible tissue damage known as coagulative necrosis [10,19].

For patients experiencing a local recurrence of prostate cancer treatment, this technique seems to be an effective treatment [10]. Nevertheless, several issues have prevented HIFU from being widely used in clinics, including (1) a lack of data comparing its effectiveness to surgery; (2) a lengthy scanning time; (3) difficulty identifying moving organs; (4) the possibility of sonic shadowing caused by the bones and gases in the bowl (as shown in Figure 3), which prevents HIFU from reaching the tumor area; and (5) a comparatively high cost [10].

### 3.2. Mechanical Effects

#### 3.2.1. Cavitation

The formation and oscillation of gas bubbles in a fluid is the basis for the cavitation phenomenon [21,22]. These gas bubbles can be of two types: endogenous or exogenous origin. Exogenous bubbles are artificial gas bubbles administered externally, whereas endogenous bubbles are small gaseous bubbles naturally occurring within cell tissues [23]. When these bubbles are subjected to ultrasound waves, which are a succession of acoustic waves with negative and positive peaks, the size of the bubbles oscillates [24].

Cavitation exists in two forms (shown in Figure 4): (1) stable cavitation and (2) inertial cavitation. The type of cavitation that occurs is determined by a number of factors, including US frequency, pressure, surface tension, and space availability. The likelihood of a cavitation event corresponds with the peak negative pressure and the ultrasound wave’s frequency. Bubbles that reach their resonance size and begin to oscillate linearly at the same frequency as the applied ultrasound wave around an equilibrium (resonance size) are said to exhibit stable cavitation [25,26]. The ultrasound wave’s frequency must match the bubble’s resonance frequency, which depends on the bubble’s radius, for stable cavitation to occur [10]. On the other hand, inertial cavitation happens at higher negative peak pressures, as it is the bubbles reacting to the stimulating ultrasound wave in a non-linear manner that causes asymmetrical oscillation and the bubble to collapse [27,28].

Numerous therapeutic uses for these biological effects exist. For example, it is possible to increase the blood-brain barrier’s permeability by applying the mechanical forces generated by stable cavitation [30,31]. Furthermore, in a process known as sonoporation, inertial cavitation can be utilized to increase an individual cell’s permeability for the transfer of genes [32]. Tumor drug delivery uses both stable and inertial cavitation [10].

The main principles associated with stable cavitation are microstreaming and Bjerknes secondary forces [10]. Due to the constant oscillation that occurs around the bubble’s resonance size during stable cavitation, a fluid flow called microstreaming occurs [33]. This microstreaming phenomenon creates strong shear forces with sufficient velocities to break surrounding particles and permeabilize surrounding tissue cells [33]. The second phenomenon caused by stable cavitation is the Bjerknes secondary forces, which are attractive and repulsive [10]. These forces can be used to attract particles close to the oscillating bubble for the shear forces to release their content [10].

Depending on where the bubble is located, inertial cavitation can have various biological effects [10]. A high-pressure, spherical, symmetric shockwave will be produced by the following explosion if the bubble bursts far from solid objects. This collapse will produce high local temperatures alongside high pressures. However, the explosion is not symmetrical and will result in a jet stream of water if the bubble collapses near a solid object, such as a cell membrane. This jet stream will strike the solid object, creating shear forces that can break particles and open pores within a cell membrane.

The likelihood of cavitation is assessed using the Mechanical Index (MI), which the FDA mandates is less than 1.9. The Mechanical index is computed using Equation (5), which employs the ultrasound wave’s frequency *f* in MHz and negative peak pressure *P^−^* in MPa.
(5)$$MI=\frac{P^{-}}{\sqrt{f}}$$

Sonoporation is one of the biological effects of cavitation, as was previously mentioned. This phenomenon, commonly used in drug delivery, describes the process by which pores form in the cell membrane when subjected to an ultrasonic wave. Sonoporation can arise from various mechanisms. For instance, pores in the cell membrane can be produced by the shearing forces and microstreaming fluid flow that stable cavitation produces, or they can be produced by the high-speed jet stream that is created when asymmetric inertial cavitation occurs near a cell membrane [34]. In addition, the bubble may be forced into the cell membrane by the ultrasonic wave’s radiation forces, which could ultimately cause the membrane to become unstable [35]. Sonoporation can happen on surfaces other than just cell membranes, like vessel walls. The size and quantity of pores formed in the vessel wall may increase if the bubble collapses close to one of the walls due to the shockwaves and microjets produced. Furthermore, it is noteworthy that the size of the pores formed by sonoporation is directly influenced by ultrasound parameters [36].

#### 3.2.2. Acoustic Radiation Forces

Acoustic radiation forces are another way ultrasonic waves can have a biological impact. The force that an ultrasound wave applies to the objects within its acoustic field is known as the acoustic radiation force. The formula for this force is *W*/*c*, where *c* is the sound’s velocity through a medium and *W* is the acoustic power. When ultrasound waves travel through a fluid, their energy is absorbed and imparted as kinetic energy to the fluid. This results in a localized fluid flow, also known as acoustic streaming [37]. The fluid viscosity *µ* and the absorption coefficient *α* can be used to calculate the flow’s velocity (*v*) using Equation (6) [38].
(6)$$\upsilon =\frac{2\alpha W}{\mu c}$$

These forces and the acoustic flows they generate cause bubbles and particles to be displaced in the medium, driving particles into the target tissue and resulting in reversible structural deformation [39,40]. Acoustic streaming can push particles against blood vessels during anticancer drug delivery, increasing drug retention time inside tumors [37], while ultrasound-induced shear forces can be used to increase the intracellular space between endothelial cells simultaneously [41]. However, blood flow in the vessels can counteract this phenomenon due to the force it exerts and its ability to prevent particles from remaining under the ultrasound wave for long periods of time [10].

### 3.3. Chemical Effects

#### 3.3.1. Free Radical Formation

The extreme temperature and pressure produced by ultrasound-induced inertial cavitation can lead to the thermal dissociation of water and the appearance of hydrogen atoms and hydroxyl radicals [42]. Sonosensitive particles are often added since they produce free radicals that enhance effects such as light emission and pyrolysis [43]. Light is generated after the bubble’s collapse as exited molecules lose their energy and free radicals rejoin. The emitted light can then activate sensitizers that create reactive oxygen species (ROS). ROS can also be produced during pyrolysis when sensitizers are chemically excited by the high temperatures produced during the inertial cavitation process. This produces free radicals, which then combine with other molecules in the aqueous medium to form ROS [44].

ROS production is influenced by a number of variables, including the kind of sensitizers utilized, the ultrasound wave’s intensity, and the frequency and potency of inertial cavitation. ROS can be produced in both extracellular and intracellular environments. ROS produced outside of a cell usually have short lifespans and minimal impact until they react with other solutes to form toxic compounds. Two categories of ROS are responsible for most chemical effects induced by ultrasound: (1) hydroxyl free radicals and (2) singlet oxygen molecular oxygen.

Hydroxyl free radicals are formed from the pyrolysis of water that occurs after inertial cavitation, and they can react with other water molecules to form hydroperoxide radicals. Furthermore, hydroxyl free radicals can modify the purine and pyrimidine bases of any DNA strands they interact with [45,46]. If enough ROS are present in a biological medium, oxidative stress on the cells can cause severe metabolic dysfunction such as peroxidation of the cell membrane lipids, cytoskeletal disruption, generation of protein radicals, modification of nucleic acid, DNA damage, and cell death [45,46].

#### 3.3.2. Endocytosis

Without the internalization process known as endocytosis, macromolecules of a sufficiently large size cannot enter cells. Endocytosis occurs through various pathways controlled by distinct intracellular molecules (such as caveolae and clathrin). Through a variety of mechanical effects, ultrasound waves can apply shear forces to cell membranes, causing endocytosis to occur [47]. When ultrasound induces alterations in a cell’s cytoskeleton, sensors impeded in the bilayer cell membrane can detect them and respond by initiating endocytosis [48]. Additionally, research shows that endocytosis is partly mediated by the production of ROS [49] and that after ultrasound was applied, sonoporation and endocytosis increased cell uptake of nanocarriers [50,51].

### 3.4. Ultrasound Effects on Vascular Tissue

There is a chance that ultrasound exposure will have vascular bioeffects [52]. While they are more frequent in therapeutic ultrasound, they are unlikely to occur at the exposure levels used in diagnostic ultrasound. Vascular effects can range widely, from increased microvascular permeability to vascular occlusion and hemorrhages [52]. The mechanical cavitation and ultrasonic thermal effects are the driving forces behind these phenomena. When cavitation occurs in the blood vessels, it can lead to gas and vapor-filled cavities that grow violently and oscillate, causing damage to the vascular tissue [52]. The pressure threshold needed to initiate cavitation-induced vascular bioeffects can be lowered by adding microbubbles (MBs).

The fundamental mechanism of the thermal effect is the tissue’s absorption of the incident ultrasound wave, which raises the temperature. Stronger hyperthermia can cause tissue ablation, but milder hyperthermia can increase microvascular permeability due to thermal effects [52]. In severe hyperthermia, microvessels experience widespread thermal coagulation [52]. This causes the vessel to collapse, which stops the blood flow. As vessels become larger, their walls become more robust, and blood flow velocities are higher. Larger vessels, therefore, require higher exposure levels to induce thermal damage. Applying ultrasound to large vessels has resulted in transient vessel spasms, which can temporarily stop blood flow [52]. Complete occlusion of vessels has also been observed [52]. The dominant cause of this occlusion is thermal coagulation of the vessel wall. HIFU is capable of stopping bleeding from lacerated vessels and tissue, such as abdominal bleeding cessation [52].

## 4. Nanoparticles Used with Ultrasound

### 4.1. Ultrasound Interaction with Nanoparticles and Drug Release Mechanisms

By combining the benefits of nanoparticle drug delivery systems (shown in Figure 5) with the biological effects of ultrasound, interactions between nanoparticles, cells, and ultrasound waves can produce a synergistic anticancer effect. Three main mechanisms define how ultrasound improves the treatment efficacy of stimuli-responsive drug delivery: (1) inducing drug release from nanocarriers at the target location, (2) improving drug and nanoparticle transport in the extracellular matrix, and (3) improving drug transport within cells once released.

#### 4.1.1. Ultrasound-Induced Drug Release

It is possible to design drug delivery systems with drug-loaded nanocarriers that break apart when exposed to the thermal and non-thermal effects of ultrasonic waves and release their payload. This will enable targeted drug delivery to the tumor, reducing the dosage needed and avoiding negative side effects in healthy tissues. It will also enable on-demand drug release that is restricted to the area of interest. Furthermore, ultrasound-mediated drug delivery systems can address the low treatment efficacy and low penetration depth of current cancer treatments.

#### 4.1.2. Improved Extracellular Transport of Drugs and Nanoparticles

The thermal and mechanical effects of ultrasound waves can improve extracellular transport of anticancer drugs and nanoparticles. Typically used to initiate thermal drug release, ultrasound-induced mild hyperthermia (40–43 °C) can improve blood flow and vessel permeability, increasing drug delivery into the target tumor’s extracellular matrix. Utilizing MBs in conjunction with ultrasound waves can also improve the transport of drugs and nanoparticles. Shear forces and microjets generated during stable and inertial cavitation may cause the drug payload within the nanoparticles to be released and disrupt the endothelium membrane of the vessel. This increases the amount of anticancer drugs delivered to the target tumor’s extracellular matrix. Ultrasound radiation forces can also push nanoparticles into the extracellular space of tumors, causing them to aggregate and penetrate more deeply.

A temporary opening of the blood-brain barrier (BBB), a semi-permeable membrane that controls molecular transport between the blood and the brain, can also be achieved by combining ultrasound waves with ultrasound-sensitizing nanoparticles. Focused ultrasound is applied to nanoparticles, such as MBs, in order to temporarily, reversibly, and noninvasively increase the blood-brain barrier’s cellular and vascular permeability [53]. This enables drug molecules that are typically restricted within the vasculature to pass through the barrier and reach the brain. Different types of nanoparticles can be combined into ultrasound-sensitizing complexes, such as chemically conjugated liposomes with MBs [54], to deliver various types of therapeutic drugs to the brain.

#### 4.1.3. Improved Cellular Drug Transport

The primary mechanisms through which ultrasound facilitates the passage of drug molecules across the cell membrane are its mechanical and chemical effects. Sonoporation is a technique that increases the permeability of cell membranes to allow different molecules to enter the cell’s cytoplasm through sonic jets and microstreaming. It is generated by the cavitation effects of ultrasound when used in conjunction with MBs. ROS, produced by the high pressures and temperatures associated with the inertial cavitation process, also impact the permeability of the cell membrane. Additionally, endocytosis can be induced by ultrasound waves applying shear forces to the cell membrane, which increases the amount of drug molecules that enter the cell’s cytoplasm.

### 4.2. Different Types of Ultrasound Sensitive Nanoparticles

Ultrasound-activated drug delivery nanocarriers fall into several main categories, such as metallic nanoparticles, liposomes, nanobubbles, exosomes, mesoporous silica nanoparticles, and nanodroplets containing perfluorocarbon. Table 2 provides some examples of organic and inorganic nanoparticles and their benefits and drawbacks. The payload drugs can be encapsulated, dissolved, or attached to the nanoparticle’s matrix or surface. The ultimate shape of the nanocarrier will be determined by its intended use [55,56]. It is also crucial to note that drug delivery systems can respond to various ultrasound stimuli, enabling more spatiotemporal control over dosage and a stepwise release of the drug [57]. This review will go over the different applications that can be made by combining each of these nanocarriers with ultrasound in the next section. A summary of some in vitro and in vivo ultrasound responsive drug delivery systems is shown in Table 3.

#### 4.2.1. Liposomes

With a size range of 20 nanometers to 1 micrometer, liposomes are concentric spherical structures typically composed of phospholipids; some also contain cholesterol molecules in their structure [78,79], which encircle an aqueous compartment. Liposomes can encapsulate hydrophilic or hydrophobic drug molecules. Hydrophobic drugs can be embedded into the liposome’s membrane, while hydrophilic drugs can be stored in the aqueous compartment [79,80]. A variety of liposome types exist, including conventional liposomes, temperature-sensitive liposomes (TSL), PH-sensitive liposomes, immunoliposomes, fusogenic liposomes, and numerous others [80,81]. The biocompatibility and cell uptake of liposome bilayers may be facilitated by their resemblance to cell membranes, which allows fusogenic liposomes to fuse with intra-cellular compartment membranes or cell membranes. However, other liposomes can be endocytosed without fusion. Liposomes can have their surfaces altered to include polymers like polyethylene glycol (PEG) chains, which aid in lowering the liposomes’ absorption by reticuloendothelial system (RES) cells [82,83]. Compared to unaltered liposomes, this resulted in longer circulation times and drug accumulation within the tumor [84].

The literature describes temperature-sensitive liposomes (TSL) as the most widely used nanocarrier with thermal stimuli [80,85]. TSLs are designed to release their payload when the area of interest is subjected to mild hyperthermia. An increase in temperature brought on by the absorption of acoustic energy in the ultrasound-exposed area can result in thermal drug release. This rise in temperature in the target region is correlated with the exposure duration, the medium’s absorption coefficient, the acoustic pressure, and the ultrasound duty cycle. In order to minimize harm, TSLs are engineered to release their contents only when the temperature within the targeted area exceeds the body’s temperature by a few degrees [86]. As an example, ThermoDox is a clinically tested thermally responsive drug delivery system that enables targeted delivery of DOX to solid tumors at temperatures higher than 40 °C [87].

Applying ultrasound waves to liposomes can cause destabilization of the liposome membrane, which can induce drug release via thermal and mechanical effects. The process of producing hyperthermia through thermal effects can lead to a transition of the lipid bilayer from a solid-order phase to a liquid-disordered phase, thereby increasing liposomal permeability and facilitating drug escape [88]. The mechanical effect of ultrasound includes the phenomenon of cavitation. When inertial cavitation generates shear forces with amplitudes larger than 10,000 atmospheres of pressure, the membrane of the surrounding liposomes may rupture, allowing encapsulated drugs to escape [88].

While hydrophilic drugs experience less drug diffusion across the liposomal membrane compared to hydrophobic drugs, hydrophobic and amphiphilic drug molecules are not properly retained in liposomes. This can cause drug leakages that can have harmful effects on healthy tissues [28]. To solve this issue, polymeric materials can surround the liposomal lipid bilayer to stop hydrophobic and amphiphilic molecules from diffusing outward. However, several benefits and disadvantages can occur depending on the polymer employed. The polymer may provide benefits such as increased targeting efficiency and prolonged circulation time. Simultaneously, it can hinder the payload’s release and the fusion of the liposome with target cells, necessitating the use of a second drug release mechanism [28].

#### 4.2.2. Polymeric Micelles

Polymeric micelles are self-assembled [89] colloidal structures made from a monolayer of amphiphilic copolymers. In contrast to liposomes, they comprise a hydrophobic core and a hydrophilic shell and are typically in the size range of 10–100 nm. Micelles are typically spherically shaped, but they can also be designed with other morphologies (such as rods or lamellae) based on the temperature and properties of the polymer’s constituent blocks [90]. The Micelle formation is most frequently facilitated by hydrophilic blocks, such as polyethylene glycol (PEG) and polyethylene oxide (PEO). Their monomer subunits (-CH2- CH2-O) are the same, but the end groups vary based on the synthesis process [91]. PEG blocks can hinder RES cells from absorbing polymeric micelles and prolong their circulation time [92]. The drug’s compatibility with the core largely determines the choice of hydrophobic blocks [91]. The hydrophobic core ensures that hydrophobic drugs are easily encapsulated and delivered to the tumor site [89]. In addition, ionic bonding and chemical conjugation can be used to add drug molecules to polymeric micelles.

Owing to the micelle structure’s chemical adaptability, numerous modifications can be made to create customized drug carriers. For example, creating cross-links between the polymer chains can increase stability against early breakdowns based on physiological stresses [91]. Moreover, the hydrophilic shell can be modified to allow for the targeted delivery of drugs by attaching ligands like antibodies, folic acid, growth factors, transferrin, or other compounds [93]. To trigger drug release, modifications may also be made to the micelles to make them responsive to various stimuli (such as heat, light, and pH decrease in the tumor environment). Not only can the more responsive micelles be used in conjunction with US heating, but micelle-based drug delivery can also benefit from the mechanical effects of US [94].

Extrinsic or intrinsic stimuli can trigger drug release from micelles once they have reached the location of interest. The disruption of the nanocarrier is necessary for the drug to be released from the micelles [95]. The mechanical effects of ultrasonic waves, such as cavitation, can produce extreme stresses and shear forces. When the shear forces generated by cavitation surpass the cohesive forces of the nanocarrier, the micelles will rupture, releasing the drug [95]. In addition, micelles combined with pulsed ultrasound can lessen the side effects of chemotherapy [95]. For example, when the ultrasound is on, the drug is released from the micelles, and when the ultrasound is off, the drugs that did not enter the cells can be re-encapsulated in the micelles, which then reform and recirculate in the bloodstream [95]. Reversible drug release can thus be produced by the micelles’ reversible disordering [96,97].

#### 4.2.3. Mesoporous Silica Nanoparticles

Mesoporous Silica Nanoparticles (MSNs) are a class of solid nanoparticles with potential applications such as sonosensitizers and cavitation nuclei when employed with ultrasound waves [98]. Solid nanoparticles are spherical in shape and typically have a solid core [79]. Because of their porous structure, which increases their surface area, MSNs are an inorganic nanocarrier with a high drug-loading capacity. They provide further benefits as nanocarriers due to their biodegradability and biocompatibility [99,100]. Molecules functioning as pore caps obstruct the MSNs’ intrinsic pores to stop the drug payload from releasing prematurely. By adding different components, MSNs can be modified to acquire more specialized functions, including longer circulation times [101], physiological stabilization [102], and targeted therapy [103]. They are therefore useful instruments for therapeutic, imaging, and drug delivery applications [104].

The loading and release effectiveness of the MSNs is determined by the drug’s solubility and interactions with the pore caps [105]. The start and duration of pore openings can be controlled by functionalizing the pore caps with different chemical groups, thereby regulating the loading and release of the drug payload. This prevents the premature release of the drug and permits its release at the region of interest [105]. Certain chemical bonds known as mechanophores, such as those found in ultrasonic-sensitive moieties, can be broken by ultrasonic waves’ mechanical and thermal effects [106]. For instance, hy drophilic methacrylic acid (MAA) can be produced by ultrasound cleaving the hydrophobic monomer 2-tetrahydropyranyl methacrylate (THPMA), which has a labile acetal group. Using ultrasound to induce phase transformation from hydrophobic to hydrophilic, a polymeric gatekeeper could act as an MSN pore cap that prevents premature drug release [106].

The combined use of ultrasound with MSNs results in a synergistic cytotoxic effect that has destroyed cancerous tumor cells in vitro [33,107]. As previously indicated, when exposed to an ultrasonic wave, MSNs function as sonosenitizers and cavitation nuclei. Their abrasive exterior causes bubbles to form inside their pores. After the dissolution of silicon nanoparticles, hydrogen bubbles are also produced [108].

#### 4.2.4. Super Magnetic Iron Oxide Nanoparticles

The diameter of superparamagnetic iron oxide nanoparticles, or SPIONs, varies from 1 to 100 nm, and they are composed of either magnetite iron oxide (Fe_3_O_4_) or its oxidized form (*γ* Fe_2_O_3_). Chemotherapy drugs can be conjugated to different functional groups and grafted directly onto SPIONs to add them to their structure. Additionally, targeting molecules like folic acid, RGD, proteins, transferrin, and hyaluronic acid can be added to the surfaces of SPIONs to improve their targeting capabilities [27]. Since SPIONs are composed of iron oxide cores, they can be directed to a specific area of interest by applying a strong magnetic field generated by an external magnet over the intended target location. The potential of SPIONs as nanocarriers for targeted drug delivery systems is enhanced by their biodegradability and ease of synthesis [109]. The size, dose, species, and surface coating of SPIONs generally determine their toxicity [110]. While in vivo research revealed varying SPION toxicity ranging from negative [111,112] to positive toxicity [113,114], some in vitro studies demonstrate that the cytotoxic effects of SPIONS on cell cultures are minimal [115,116,117,118].

SPIONs were identified as ROS-producing sonosensitizers [119]. When SPIONs are exposed to ultrasound (1 MHz frequency and 2 W/cm^2^), there are increased levels of cellular ROS, which in turn cause a rise in cell death [119]. Researchers found that a viable strategy to boost the production of ROS brought on by ultrasound is to coat SPIONs with sonosensitizer molecules [120].

Because the drugs are only loaded onto the nanoparticle’s surface, early drug release has been identified as a problem when using SPIONs as nanocarriers. This can prevent chemotherapy drugs from reaching the target area in sufficient concentrations. To control this problem, the metal cores are coated in aqueous solutions with biocompatible polymers. Additionally, these polymers enable drug binding by covalent bonding, adsorption, or particle entrapment while safeguarding the magnetic core [26].

#### 4.2.5. Gold Nanoparticles

Although Gold Nanoparticles can be used as anticancer therapeutic agents on their own, they can also be used in drug delivery applications. Gold nanoparticles (GNPs) are inorganic nanoparticles with unique characteristics that render them promising nanocarriers for chemotherapy drugs. These properties include energy absorption, size (10–100 nm), and stability. Chemotherapy drugs and GNPS can conjugate to form drug-delivery nanocarriers [121]. Alkanethiol linkers and physical adsorption are two ways conjugation can happen [122]. Additionally, PEG polymers can be attached to GNPs to prevent RES cells from removing them, and ligands like transferrin and folic acid can be attached to GNPs to make them target tumor cells [53,123]. It was discovered by researchers [124] that a receptor-mediated, clathrin-dependent endocytosis pathway mediates the cellular uptake of gold nanoparticles. Comparing GNPs to larger nanocarriers, they found that GNPs’ smaller size results in faster uptake and higher concentrations inside the cells.

Using a phase-changing substance as the drug-loading medium, hydrophobic and hydrophilic medications have been loaded into gold nanoparticles [125]. The phase-changing material will not release the drug until it reaches its melting point. HIFU can be applied to the gold nanoparticles to increase their temperature to initiate the phase change and allow the drug to diffuse out.

When GNPS are present in an aqueous medium, their rough surface provides bubble nuclei that lower the ultrasound pressure threshold needed for cavitation [126]. Using GNPs with ultrasound waves has been shown to significantly reduce cancer cell growth and spread [127], increase ROS generation [127], and improve inertial cavitation [128].

#### 4.2.6. Microbubbles

Originally created as ultrasound contrast agents, MBs have been considered for use in ultrasound-mediated drug delivery [129]. Their usual composition consists of a gaseous core (such as a mixture of perfluorocarbon (PFC) gas and air) encased in a 2–500 nm-thick protein, lipid, surfactant, or polymer shell. The shell’s composition affects MBs stiffness, resistance to ultrasound-induced rupture, and clearance by RES cells [130]. The perfluorocarbon gas (PFC) density in the microbubble’s gaseous core differs considerably from that of the surrounding medium. This enables the easy manipulation of MBs via ultrasound-induced acoustic radiation forces and increases the microbubble’s sensitivity to the pressure produced by ultrasonic waves [131].

One of the most popular surfactants for coating MBs is phospholipids. Phospholipids can naturally form a monolayer around the gaseous core due to their polar head and two hydrophobic tails. The phospholipid shell can then be altered to enable the attachment of drugs to the shell through covalent or non-covalent bonding or to allow the incorporation of hydrophobic drug molecules in the hydrophobic shell. Indeed, it is possible to encapsulate or attach drug-loaded nanoparticles to the surface of the microbubble shell [132]. An additional method of microbubble drug loading involves incorporating an oil phase that includes drugs into the microbubble [133].

There is promising potential for enhanced drug delivery when combining ultrasound with MBs. When exposed to ultrasonic waves, MBs undergo cavitation, acting as vibrating bubbles. Focused ultrasound can be used to trigger the microbubbles to burst, releasing the drug. Cell membrane permeabilization and drug release can occur simultaneously when ultrasound waves are applied to drug-loaded MBs. The delivery of drugs can be guided and tracked through the use of low-intensity ultrasound to image the MBs [134]. Additionally, MBs can be combined with other nanocarriers to improve targeted drug delivery through the use of ultrasound-triggered microbubble destruction [135,136].

Even though the FDA has approved MBs for use in humans, they still have certain drawbacks. Several characteristics of MBs, such as their relatively large micrometer size, short circulation time, and restricted drug loading capacity, pose challenges to the successful delivery of drugs [137]. Another aspect to consider is that injecting MBs into blood vessels can have unfavorable side effects, such as blood vessel dilatation and elevated osmotic blood pressure [138,139].

Converting MBs into nanobubbles (size range of 5–500 nm) is one potential way to get around some of the limitations and drawbacks of MBs [140]. Even though there are certain drawbacks to nanobubbles, like low echogenicity and instability, these can be worked around by adding pluronic acid to the nanobubbles. In comparison to MBs, pluronic-modified nanobubbles demonstrated increased stability, longer circulation times, and improved echogenicity [141]. While nanobubbles have many advantages, including enhanced drug delivery into tumors, triggered drug release, and real-time visualization, they also face obstacles, such as cellular internalization, escape from endosomal entrapment, and limited drug load capacity [142].

#### 4.2.7. Exosomes

Exosomes are 40–120 nm in size. Phospholipid bilayer vesicles can be produced by a wide variety of cells, including B cells, T cells, dendritic cells, macrophages, neurons, glial cells, and most tumor cell lines [143]. Because of their inherent qualities, exosomes are being investigated as possible drug and gene-delivery vehicles [143]. By transferring genetic material between cells in an organism’s natural pathway, they play a crucial role in intercellular communications. Being naturally occurring cell-to-cell transporters, exosomes can contain many materials, including proteins, lipids, DNA, and RNA that can regulate the recipient cells. There is growing evidence that exosomes combine the benefits of cell-mediated delivery systems and synthetic nanocarriers [143].

Exosomes are useful as nanocarriers in drug delivery systems for a number of reasons. They are capable of remaining in the blood vessel for extended periods [143]. They are able to encapsulate soluble drug molecules because of their hydrophilic core [143]. Furthermore, they are also capable of overcoming a number of biological barriers [143]. Additionally, because of the cell surface molecules they carry and their nano-scale size, they have a natural targeting ability that reduces off-target effects [143].

Exosome drug loading has been found to be effective when ultrasound waves are applied to a drug exosome mixture [143]. The drug-loaded exosomes stayed stable for months under a variety of conditions. Additionally, ultrasound applications can increase the drug delivery efficiency of exosomes [131,144].

## 5. Challenges Facing Ultrasound Responsive Nanocarriers in Cancer Treatment

Only a small number of nanocarriers are commercially available, despite these materials’ promising potential. Overcoming some of the technology’s drawbacks, including large-scale production, safety concerns, and successful reproducibility, is essential for the future of nanocarriers [145]. Standard criteria for toxicity assessment are necessary since it is still challenging to assess the toxicity of nanomaterials and develop validated advanced complementary assays [146,147]. Owing to the distinct qualities of nanocarriers, standard drug assays might not be adequate for fully evaluating the toxicity of nanoparticles. Furthermore, a uniform list of necessary tests is absent.

### 5.1. Challenges Associated with Nanoparticle Toxicity

The most crucial factors in determining the toxicity of nanocarriers appear to be size and surface charge. The relationship between toxicity and nanocarrier size is inverse; the greater the toxicity, the smaller the nanoparticle size, and vice versa [148,149]. Larger nanoparticles tend to accumulate in the liver and spleen of the mononuclear phagocyte system, whereas smaller nanoparticles (10–15 nm) have a broad biodistribution [150]. Since highly cytotoxic drugs may be contained in cancer treatment nanocarriers, the nanocarrier must exhibit some stability to avoid premature drug release that could harm healthy tissues. Consequently, a shell surrounding nanocarriers is crucial for (reducing) the toxicity of nanocarriers [151]. Additionally, the route of administration affects the toxicity of nanoparticles, since their biodistribution and toxicokinetics are altered depending on the exposure route [150]. Since the exposure route affects the biodistribution and toxicokinetics of nanoparticles, the route of administration also influences the toxicity of the particles [150].

According to Roman Lehner et al. [152], repeated exposure was shown to cause hypersensitivity reactions, increased clearance rates, immune system activation, and the formation of antibodies against PEG. The biosafety of PEGylated drug formations has become questionable. However, these side effects may be tolerated in patients with serious medical conditions [152].

Another crucial biosafety issue is the application of ultrasound. A number of ultrasound parameters, including frequency, focusing, pulse repetition frequency, pulse duration, exposure time, and intensity, as well as the attenuation coefficient and acoustic impedance of biological tissues, determine the extent and severity of thermal and mechanical ultrasound effects. The mechanical and thermal effects that are beneficial for treating cancer may also have negative biological effects on healthy tissues [153].

Since ultrasound applications can cause unwanted cavitation effects in the presence of air bubbles, organs such as the lungs and bowels are not appropriate for US treatment. The thermal (TI) and mechanical (MI) indices were created to reduce the possibility of thermal or mechanical injuries brought on by ultrasound. In light of the biological effects of ultrasound, these indices are consequently useful for developing safe and efficient techniques for tumor-specific imaging and therapy [10].

### 5.2. Challenges Associated with Clinical Adoption of Nanoparticles

Bringing nanomedicine to market presents several challenges. Clinical translation requires scaling up nanoparticle manufacturing to an industrial level with high yield and reproducibility. This process is difficult because small nanoparticle batches are easier to control and optimize, whereas maintaining quality control in large-scale production is both complex and expensive. As a result, investors are often hesitant to enter the nanomedicine field, creating further obstacles to its clinical adoption. The costs associated with developing nano-formulations from the lab to clinical use are prohibitive, with success only emerging after years of investment. Additionally, investors’ reluctance is reinforced by the cost-benefit analysis of nanomedicine.

Further research is needed to deepen our understanding of human biology and the interactions between nanoparticles and biological systems, which will increase the likelihood of nanomedicine’s clinical adoption. A better understanding of factors such as tumor heterogeneity and the effects of nanoparticle size, shape, or coating on toxicity, dispersion, and aggregation is essential. Thorough characterization of nanoparticles used in drug delivery systems is also necessary. Both in vitro and in vivo studies can be employed for this purpose. While in vitro studies can provide proof of concept for drug delivery systems and their nanocarriers, in vivo studies are crucial for modeling the nanoparticles’ properties, pathways, and potential intercellular interactions.

## 6. Conclusions

The primary aim of this review was to explore the potential applications of ultrasound waves in combination with various nanocarriers for cancer treatment. When compared to traditional chemotherapy, ultrasound-induced drug delivery systems offer several advantages. These effective and adaptable techniques improve treatment efficacy while reducing unwanted side effects. In ultrasound-mediated drug delivery, drugs are typically encapsulated in nanocarriers, including liposomes, polymeric micelles, mesoporous silica nanoparticles (MSNs), superparamagnetic iron oxide nanoparticles (SPIONs), MBs, nanobubbles (NBs), and exosomes. These nanocarriers are designed to be responsive to ultrasound’s thermal and mechanical effects, making them sensitive to increased temperatures and pressure.

Once the drug molecules are encapsulated, the drug-loaded nanocarriers are administered intravenously. This encapsulation minimizes the toxicity to healthy cells by preventing premature drug release. External ultrasound waves are then used to trigger the release of the drug payload. The release can be spatially and temporally controlled by focusing and adjusting the ultrasound waves, ensuring the drug is delivered precisely at the tumor site at the right time. Without ultrasound, the drug-loaded nanocarrier can circulate harmlessly through the body.

However, several challenges must be addressed to ensure the success of ultrasound-mediated drug delivery in clinical settings. Both in vitro and in vivo studies are necessary to assess and mitigate the potential toxicity of the nanocarriers. Multidisciplinary research is also required to optimize ultrasound parameters, equipment, and ultrasound-responsive nanocarriers. The ultimate goal of this research should be to maximize treatment efficiency by developing practical devices that ensure patient safety and allow for real-time monitoring of ultrasound’s thermal, mechanical, and chemical effects.

## Figures and Tables

**Figure 1 pharmaceutics-16-01383-f001:**
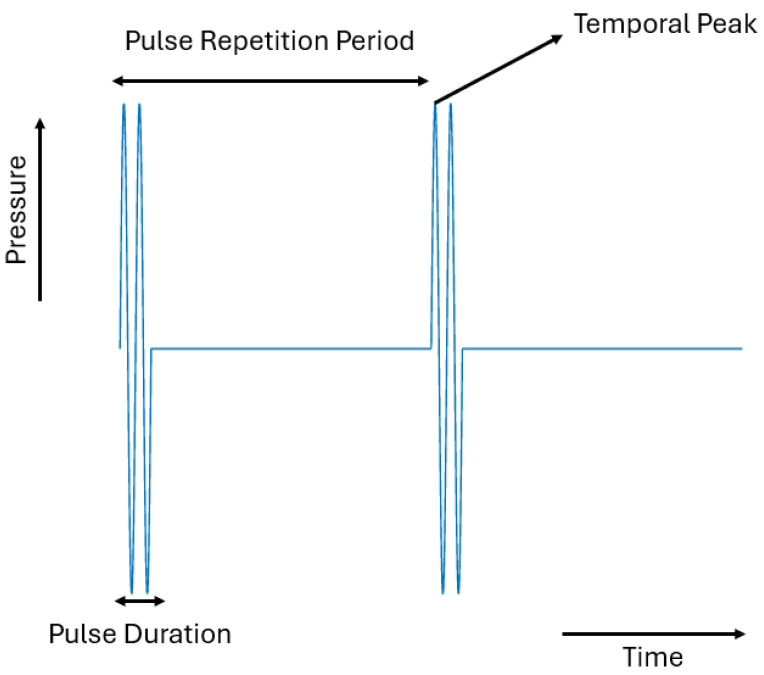
The Three Temporal Characteristics of a Discontinuous Ultrasound Wave.

**Figure 2 pharmaceutics-16-01383-f002:**
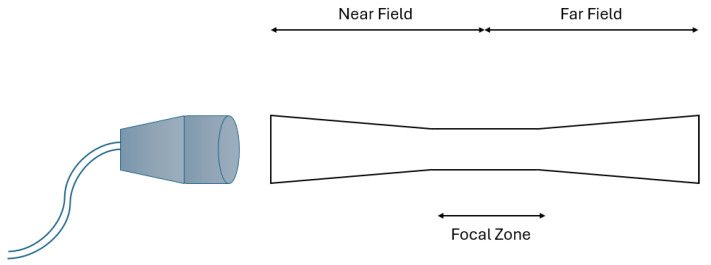
The shape and the three sections of an ultrasound beam (adapted from reference [12]).

**Figure 3 pharmaceutics-16-01383-f003:**
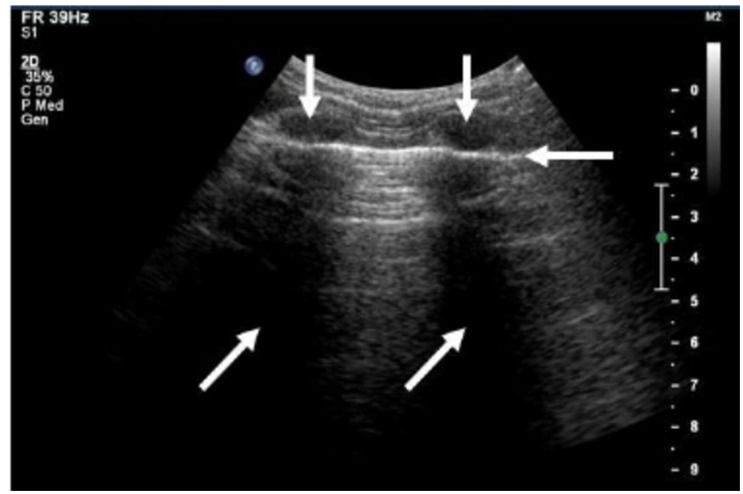
Sonic shadows produced by two adjacent bones in the rib cage. The ultrasound image shows two neighboring ribs (vertical arrows), a hyperechogenic pleural line (horizontal arrow), and acoustic shadows below (oblique arrows) (Retrieved with open access permission from reference [20]).

**Figure 4 pharmaceutics-16-01383-f004:**
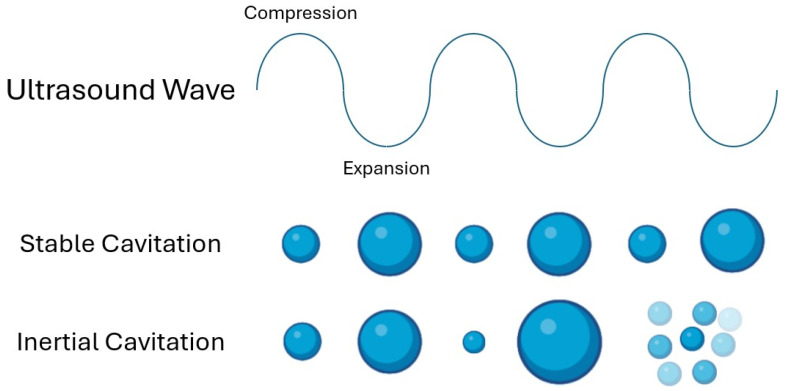
Stable and inertial cavitation. Stable cavitation shows repetitive pulses around an equilibrium. It is characterized by larger bubbles and lower pressures. Inertial cavitation shows unstable growth and a violent collapse. It is characterized by smaller bubbles and higher pressures. Image created using the www.biorender.com (Adapted from Reference [29]).

**Figure 5 pharmaceutics-16-01383-f005:**
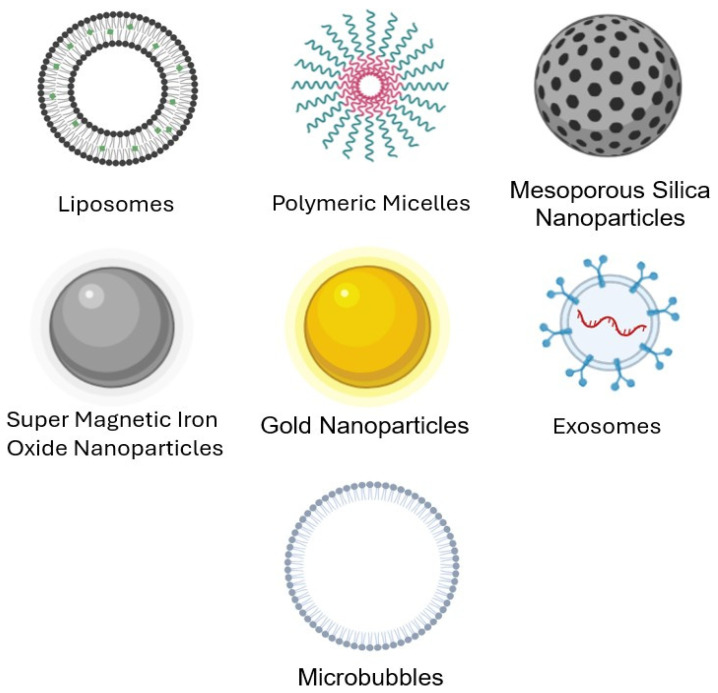
Ultrasound responsive nanocarriers can either be made of organic or inorganic materials. This figure shows the types of organic and inorganic nanoparticles discussed in this review (Image created using the Biorender Web Application (www.biorender.com)).

**Table 1 pharmaceutics-16-01383-t001:** The advantages and disadvantages of drug delivery trigger mechanisms.

Triggering Mechanism	Type	Advantages	Disadvantages
Magnetic Field	Extrinsic	Non-ionizing radiation, energy modulation using alternating magnetic fields, possible imaging opportunities	Particle accumulation can result in toxicity, limited to surface tumors, expensive, non-mobile equipment
X-Ray	Extrinsic	High precision, high tissue penetration, easily tuned	Ionizing radiation, expensive, non-mobile equipment
Microwaves	Extrinsic	Non-ionizing radiation, Non-invasive, easily tuned	Low tissue penetration, possible increase in temperature
Light	Extrinsic	Non-ionizing radiation, non-invasive	Low tissue penetration
Ultrasound	Extrinsic	Cost, easily accessible, easily tuned, high spatial and temporal precision, mobile equipment, possible imaging utility	Difficulty in application on moving objects and large volumes
pH Difference	Intrinsic	Targets low PH tumor environment, wide applicability, simple structure	PH-sensitive drug delivery systems have poor site specificity and can cause off-target delivery, difficulty in maintaining structure during drug delivery
Redox Reactions	Intrinsic	Able to target a tumor environment or disease site because of variations in redox potential as a result of molecules like GSH being present	Careful design is required to ensure specificity, performance may be impacted by the body’s variable redox environments.
Enzymatic	Intrinsic	High specificity for environments overexpressing certain enzymes, potential for minimal side effects	Environmentally sensitive, unpredictable in vivo response, difficulty in designing enzyme-responsive drug delivery system

**Table 2 pharmaceutics-16-01383-t002:** The advantages of some examples of organic and inorganic nanoparticles.

Nanoparticles	Type	Advantages	Disadvantages
Liposomes	Organic	Easy preparation, good biocompatibility, low toxicity, enhanced circulation time through pegylation, able to encapsulate both polar and non-polar molecules, functional groups can be added for targeted drug delivery	Limited storage conditions, low stability, potential allergic reactions
Dendrimers	Organic	High drug loading capacity, functional groups can be attached to the outer surface	Complex and costly synthesis process, possible toxicity, limited solubility for hydrophilic molecules
Polymeric Micelles	Organic	Self-assembly and chemical flexibility allow for modification, good stability, biodegradable, and biocompatible	Toxic organic solvent residue leftover from the formation process can affect aggregation properties, low drug loading capacity, challenges in industrial scale production
Exosome	Organic (Biological)	Natural nanocarriers, intrinsic targeting that prevents off-target effects, can pass through biological barriers and reduce Immune response, low to moderate starting material cost, enhanced efficacy and pharmacokinetic profile, and low toxicity	Undesired effects due to exosome components, lack of standardized production method
Solid Lipid Nanoparticles	Organic	A high surface area to volume ratio allows for high rug loading capacity, high stability, biocompatible, functional groups can be conjugated for active targeting	Difficulties in large-scale reproducible synthesis, stability issues, low drug loading capacity
Gold Nanoparticles	Inorganic	It can be directed using external magnetic fields	Potential toxicity depends on nanocarriers size, shape, and surface modification, in vitro studies have shown it can induce ROS production leading to DNA damage, and cell death, in vivo studies are required to fully assess the toxicity, size, dose, species, and surface coating
Super Magnetic Iron Oxide Nanoparticles	Inorganic	Ease of production, loss of magnetism in the absence of magnetic field lowers risk of particle accumulation, ROS-producing sonosensitizers stable, tunable, and uniform pore size, controlled	determine their toxicity, risks associated with inhalation, ingestion, and skin absorption, synthesis challenges, possible toxicity depending on the size
Nanoparticles	Inorganic	Release of drug payload, high drug loading capacity because of their porous structure	Potential toxicity affected by the shape, size, surface functionality, hydrophilicity, porosity, and surface conductivity
Carbon Nanotubes	Inorganic	Large surface area, sustained release while safeguarding the entrapped drug, possible surface modification	Poor water solubility
Quantum Dots	Inorganic	Imaging properties, theranostic utility, control of particle size and surface charge of the nanoparticle, smaller particles can penetrate cell membranes easily	Highly toxicity due to their composition, possible particle accumulation, instability due to air sensitivity and possible oxidation

**Table 3 pharmaceutics-16-01383-t003:** Summary of important in vitro and in vivo studies on ultrasound trigger drug delivery systems.

Reference	Study Type	U.S. Parameters	Nanoparticle Type/ Drug Type	Effect
[58]	in vitro	20 to 90 kHz, 0 to 3 W/cm^2^	Pluronic Micelles/ DOX and Ruboxyl	When the micelles were ruptured by cavitation, the encapsulated drug was released. The release of DOX was greater than that of ruboxyl.
[59]	in vitro	20 kHz (1.4, 14, and 33 mW/cm^2^	Pluronic Micelles/DOX	The process of sonication increased the uptake of DOX by cancerous cells.
[60]	in vitro	1, 3 MHz and 20 kHz 3 MHz power densities (0.058, 6 and 0–0.2 W/cm^2^)	Micelles/DOX	Drug release from micelles and intracellular drug uptake by cancer cells are both increased by sonication
[61]	in vitro	1.1 MHz (0–150 W)	(PPG-[Cu]-PEG) micelles/(pyrene/Nile red).	After HIFU sonication, the encapsulated payload was rapidly released from the micelles.
[62]	in vitro	1.1 MHz (0–150 W)	copolymer micelles/ (pyrene/DTT)	Redox and HIFU combined to improve drug release from copolymer micelles.
[63]	in vitro	1 MHz, 2 W/cm^2^	Liposome-loaded (lipid-shelled) MBs/DOX	US-mediated drug release; even at low DOX doses, cancer cells are killed
[64]	in vitro	70 kHz	Polymeric Micelles /(DPH/DOX)	DOX release increased with temperature from 25 °C (2%) to 37 °C (4%). Stopping the sonication led to DOX re-encapsulation
[65]	in vitro	1 MHz	Thermosensitive Liposomes/Calcein	TSL drug release was improved by the focused US due to the mechanical stresses that were produced.
[66]	in vitro	20 kHz (1 W/cm^2^)	eLiposomes/DOX	Compared to liposomes without emulsions eLiposomes showed increased DOX release after sonication.
[67]	in vitro	1.5 MHz and 35 mW/cm^2^ for 10–80 min	MBs/basic fibroblast growth factor (bFGF)	Without obvious cytotoxicity, greatly increased the efficiency of bFGF, cellular uptake, and flow cytometry to MI tissue.
[68]	in vivo	1.0 MHz; 3 min; TAT 3 W; 30% duty cycle	DVDMS liposomes conjugated to MBs	MBs and DVDMS sonsensitizer sonification helped reduce the size of the tumor
[69]	in vivo	3 MHz, 3.1 W	Micelles and Nanoemulsions/Paclitaxel (IV)	Compared to micelles with solid cores, those with elastic cores and the corresponding nanoemulsions showed higher treatment efficacy. Nanoemulsions showed less systemic toxicity compared to micelles.
[70]	in vivo	1.54 MHz, pulsed	CuDOX-TSL/DOX	Approximately 100% tumor inhibition
[71]	in vivo	1 MHz	TSL/DOX	The TSL liposomes in conjunction with HIFU dramatically reduced tumor regression.
[72]	in vivo	1.7 MHz	Liposomes/DOX	Liposomal DOX’s therapeutic effect in the brain was enhanced by the US-mediated disruption of the BBB.
[73]	in vivo	1 MHz (2.9 W/cm^2^)	Liposomes/DOX	Improved targeted drug delivery brought about by ultrasound application inhibited the growth of brain tumors.
[74]	in vivo	4 W/cm^2^	Mesoporous Silica Nanoparticles/DOX	High drug-loading properties and synergistic effects between ultrasound and drug delivery system
[75]	in vivo	1 MHz, 2 W/cm^2^	Microbubble-encased Mesoporous Silica nanoparticles/TAN	High drug loading capacity and multitargeting capability.
[76]	in vivo	1.1 MHz	Liposomes with Microbubble/PTX	When liposomes and US were combined with MBs, the drug’s efficacy was increased.
[77]	in vivo	MI 1.3, peak negative pressure 2.3 MPa, 50% duty cycle, 3.13 MHz frequency, and 80 mW acoustic power	liposome–microbubble complexes/Ganoderma applanatum polysaccharide	Reduced the growth of rabbit VX2 liver tumors in the SH field by blocking TAMs

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
