# Peer review of "Drug Release via Ultrasound-Activated Nanocarriers for Cancer Treatment: A Review"

_pharmaceutics, 2024, doi:10.3390/pharmaceutics16111383_

Round 1
Reviewer 1 Report
Comments and Suggestions for Authors
The article is a revew paper to describe physics and biological effects of ultrasound, and nanoparticles of drug delivery/release for cancer treatment. The major concerns include the unclear correlation between ultrasonic parameters and delivery/release of each type of nanoparticle, as well as the citation of overly early references. Here are some minor comments.
1. Page 3, Line 101: Delete the square brackets.
2. Page 4, Liine 110: Remove "109".
3. The equation 2 is not correct. Confirm it and correct it.
4. Page 7, Line 209: Cite the references for the intensity range from 0.125 to 3 W per centimeter squared.
5. Typo: Bjerkenes --> Bjerknes.
6. Reference 37 does not describe the Equation 6. Please cite the correct reference.
7. Page 10, Line 333: Please correct 'fluid velocity' to 'fluid viscosity'.
8. Page 16, Line 443: (Shown in ??) Add the information or remove it.
9. Page 16, Line 493: [?] a typo?
Reviewer 2 Report
Comments and Suggestions for Authors
This manuscript provides a general concise authoritative review on the use of nanocarriers in ultrasound-mediated drug delivery systems for cancer treatment. The topic is highly relevant and addresses important gap sin cancer treatment, i.e. to improve the efficacy and safety of chemotherapy through novel delivery systems. The manuscript emphasizes the novel advantages of ultrasound-mediated drug delivery over, and in comparison, with traditional chemotherapy. The written content is generally well-structured and informative stating novelty and including updated references.
Major: Some issues that could be improved: The manuscript could benefit from a clearer explanation about the mechanisms by which ultrasound could possibly induce drug release from the nanocarriers. It would be interesting to know what makes the nanocarriers disrupt and release their content.
It important to discuss if ultrasound causes disruption of the vasculature in tissues both with respect to immediate effects and on longer scale (cell death, inflammation etc). There is not much specification concerning the particular strategy of UL-particles degradation within particular tissues, e.g. in brain, the use of UL for providing drug delivery has long a site of controversy. Perhaps the author could discuss more about how the blood brain barrier is disruption to allow drug to enter the brain, and if this causes immediate effects.
Minor: This sentence (or at least its content): "Conventional chemotherapy is still used...but the drugs are not very selective and kill both cancerous and healthy cells" is being used repetitively, perhaps the authors could revise and reconsider relevant passages.
BBB: Define BBB (since it is only mentioned once the authors may as well just writ the full names).
Comments on the Quality of English LanguageIt is good
Reviewer 3 Report
Comments and Suggestions for Authors
The article entitled Drug Release Via Ultrasound-Activated Nanocarriers for Cancer Treatment: A Review by Refaai et al. is a well-written manuscript that describes the relevance of targeted drug release in cancers.The article is up-to-date and summarizes the importance of the ultrasonic methods and nanocarriers. The paper is well-organized, moving from a general introduction to more specific details about ultrasound physics, biological effects, and the different types of nanoparticles used.
The tables provide useful summaries, but some entries should be improved (e.g., “potential toxicity” under gold nanoparticles). Could you include more detail about what is currently known regarding the clinical use of these materials and what specific toxicity concerns remain?
Continue to improve the clarity of visual aids. While most figures are helpful, some could use more explanatory notes (e.g., Figure 4 on cavitation could better describe the visual differences between stable and inertial cavitation).
Please, check the inconsistencies in the References section. The journal names should be written with their abbreviations.
There are a very low number of self-citation with 3 articles in the refernces.
Comments on the Quality of English Language
The language in the manuscript is generally good, but can be improved by simplifying complex sentences, using active voice, and ensuring consistent technical terminology. Occasional grammatical issues and word choice could benefit from revision to improve scientific clarity. Lastly, ensuring consistent tense and reducing redundancy would enhance overall readability and flow.
